# Autoprobiotics in the Treatment of Patients with Colorectal Cancer in the Early Postoperative Period

**DOI:** 10.3390/microorganisms12050980

**Published:** 2024-05-13

**Authors:** Elena Ermolenko, Natalia Baryshnikova, Galina Alekhina, Alexander Zakharenko, Oleg Ten, Victor Kashchenko, Nadezhda Novikova, Olga Gushchina, Timofey Ovchinnikov, Anastasia Morozova, Anastasia Ilina, Alena Karaseva, Anna Tsapieva, Nikita Gladyshev, Alexander Dmitriev, Alexander Suvorov

**Affiliations:** 1Scientific and Educational Center “Molecular Bases of Interaction of Microorganisms and Human”, World-Class Research Center “Center for Personalized Medicine”, Institute of Experimental Medicine, 197376 St-Petersburg, Russia; lermolenko1@yandex.ru (E.E.); aomorozova1993@gmail.com (A.M.); anna.tsapieva@gmail.com (A.T.); krinege@mail.ru (N.G.);; 2Department of Internal Disease of Stomatology Faculty, Pavlov First St-Petersburg State Medical University, 197022 St-Petersburg, Russia; 3Laboratory of Medico-Social Problems of Pediatry, St-Petersburg State Pediatric Medical University, 194100 St-Petersburg, Russia; 4Oncology Department, Pavlov First St-Petersburg State Medical University, 197022 St-Petersburg, Russia; 9516183@mail.ru; 5North-Western District Scientific and Clinical Center Named after L. G. Sokolov, 194291 St-Petersburg, Russiagushina@mail.ru (O.G.);; 6Department of Faculty Surgery, St-Petersburg State University, 199034 St-Petersburg, Russia; surg122@yandex.ru; 7Beloostrov High Technology Clinic (MMC VT LLC), 188652 Leningrad Region, Russia; 8Microbiology Department, St-Petersburg State University, 199034 St-Petersburg, Russia; 9Department of Molecular Biotechnology, Saint-Petersburg State Institute of Technology, 190013 St-Petersburg, Russia

**Keywords:** microbiome, autoprobiotics, *Enterococcus* spp., *Parvomonas micra*, alpha diversity, gut dysbiosis, interleukins

## Abstract

Despite great advances in the treatment of oncological diseases, the development of medical technologies to prevent or reduce complications of therapy, in particular, those associated with surgery and the introduction of antibiotics, remains relevant. The aim of this study is to evaluate the effectiveness of the use of autoprobiotics based on indigenous non-pathogenic strains of *Enterococcus faecium* and *Enterococcus hirae* as a personalized functional food product (PFFP) in the complex therapy of colorectal cancer (CRC) in the early postoperative period. A total of 36 patients diagnosed with CRC were enrolled in the study. Study group A comprised 24 CRC patients who received autoprobiotic therapy in the early postoperative period, while the control group C included 12 CRC patients without autoprobiotic therapy. Prior to surgery and between days 14 and 16 post-surgery, comprehensive evaluations were conducted on all patients, encompassing the following: stool and gastroenterological complaints analysis, examination of the gut microbiota (bacteriological study, quantitative polymerase chain reaction, metagenome analysis), and analysis of interleukins in the serum. Results: The use of autoprobiotics led to a decrease in dyspeptic complaints after surgery. It was also associated with the absence of postoperative complications, did not cause any side effects, and led to a decrease in the level of pro-inflammatory cytokines (IL-6 and IL-18) in the blood serum. The use of autoprobiotics led to positive changes in the structure of escherichia and enterococci populations, the elimination of *Parvomonas micra* and *Fusobacterium nucleatum*, and a decrease in the quantitative content of *Clostridium perfringens* and *Akkermansia muciniphila*. Metagenomic analysis (16S rRNA) revealed an increase in alpha diversity. Conclusion: The introduction of autoprobiotics in the postoperative period is a highly effective and safe approach in the complex treatment of CRC. Future studies will allow the discovery of additional fine mechanisms of autoprobiotic therapy and its impact on the digestive, immune, endocrine, and neural systems.

## 1. Introduction

The microbiota of the human body is considered an extracorporeal organ, exerting a diverse array of functions, notably contributing to immune responsiveness and actively engaging in the functioning of nearly all physiological systems, encompassing the digestive, immune, and central nervous systems, as well as metabolic processes [1]. A discernible correlation has been established between the compositional makeup of the gut microbiota and the onset of cancer, particularly colorectal cancer (CRC). Qualitative and quantitative perturbations in the gut microbiota are intricately linked to manifestations of immunological imbalance, typically evidenced by the suppression of innate and humoral immunity, proliferation of lymphoid tissue, and decoupling of growth limitation and tumor tissue lysis mechanisms [2,3,4,5].

According to the literature, notable alterations in the gut microbiota are universally experienced by all patients with CRC. Despite the somewhat conflicting data revealed in a meta-analysis concerning the microbiota and metabolome composition in CRC, a prevalent pattern emerges, characterized by a reduction in biological diversity and an elevation in the abundance of specific taxa. Notably, an increased representation of genera such as *Fusobacterium*, *Peptostreptococcus*, *Bacteroides*, *Eubacterium*, *Prevotella*, *Clostridium*, *Campylobacter*, and members of the family *Enterobacteriaceae* is frequently observed [6,7]. Furthermore, a significant correlation is noted between CRC and an augmented abundance of specific bacterial species, including *Bacteroides fragilis*, *Fusobacterium nucleatum*, *Streptococcus spp.*, *Parvimonas micra* and *Peptostreptococcus stomatis*, *Enterobacter* spp. [6,8,9]. A significant role in the genesis of CRC is played by pathogenic *Escherichia coli* [10], *Enterococcus faecalis* [11], mucolytic bacteria *Akkermansia muciniphila* [12], and inducers of cholesterol synthesis *Peptostreptococcus anaerobius*, stimulating the proliferation of colonocytes [13]. Conversely, bacteria belonging to the genera *Collinsella*, *Slackia*, *Faecalibacterium*, and *Roseburia* exhibit an anti-oncogenic effect [14,15,16,17,18].

Surgical intervention, coupled with the administration of antibacterial drugs, typically results in more pronounced consequences, as evidenced by significant alterations in clinical and laboratory parameters, along with shifts in microbiota composition. These changes set the stage for severe complications, including but not limited to abdominal sepsis, pseudomembranous colitis, antibiotic-associated dysbiosis, and colitis [19,20].

Probiotic bacteria, such as *Streptococcus faecalis*, *Clostridium butyricum*, *Bacillus mesentericus*, *Lactobacillus plantarum 299v*, *L. plantarum*, *L. casei*, *Bifidobacterium* spp. (particularly, *B. longum*), and *Saccharomyces cerevisiae* have demonstrated mechanisms of antitumor protection [19,21,22,23,24,25]. The anticarcinogenic efficacy of probiotics is associated with the inhibition of pathogenic bacteria colonization on the gut mucosa, reinforcement of barrier functions, stimulation of mucin production, and the expression of tight junction proteins. Additionally, they foster “homeostatic” immune responses by enhancing the proliferation of Treg cells, regulating the production of pro-inflammatory cytokines and increasing apoptosis in cancer cells [26].

According to the literature data, the intake of probiotics is recommended in the complex treatment of CRC patients across all stages, encompassing the perioperative and long-term postoperative periods, as well as during and after chemoradiotherapy. Traditional therapeutic modalities, including surgery and chemoradiotherapy, exert deleterious effects on the body, notably contributing to the exacerbation of gut dysbiosis [19,20,27]. Nevertheless, in instances where probiotics exhibit insufficient efficacy, rapid elimination from the gut, or lead to side effects such as acidosis, dyspepsia, and infectious complications [28], alternative methods for rectifying clinical and laboratory parameters in CRC have been found.

Our research center (The Federal State Budgetary Institution “Institute of experimental medicine”) has devised a medical technology enabling the isolation of indigenous beneficial bacteria (e.g., lactobacilli, bifidobacteria, or enterococci) from fecal samples for gut microbiota restoration in patients with different diseases [29]. The application of autoprobiotics as a personalized functional food product (PFFP) represents an innovative therapeutic approach with several advantages over the use of commercial probiotics. Key advantages of employing indigenous non-pathogenic strains of lactobacilli, bifidobacteria, and enterococci include biological compatibility with other components of the host microbiota, adaptation to the body’s living conditions, minimal immune system burden, and positive effects on digestion [30].

Enterococcal strains have been used as probiotics in many countries. Examples include Linex (LEK, Slovenia), Bifiform (Ferrosan, Denmark), Symbioflor 1 (Symbio Pharm, Germany), and Laminolakt (Avena, Russia) [30,31]. Enterococci being the common part of indigenous human microbiota, numerous fermented food products and probiotics presently are often positioned as health-threatening bacterial pathogens. However, the difference between the clinical strains and strains used as probiotics is more than significant. The efficacy and safety of probiotic enterococci are widely investigated [32,33,34].

We chose *Enterococcus* spp. for two important reasons: (1) Enterococci belong to the family of lactic acid bacteria colonizing both the large and small intestines of the human body and can be found in everybody’s microbiota. (2) These bacteria were more effective in our previous experiments, when comparing the effects of indigenous enterococci, lactobacilli, bifidobacteria, and their mixtures on models of experimental dysbiosis and on the cell culture [35,36]. In addition, these bacteria can be easily cultivated and do not die as quickly as bifidobacteria or lactobacilli in the presence of oxygen. Genetic studies of enterococci revealed that probiotic strains selected for human consumption are quite different from clinical isolates by the organization of their genomes, the presence (or absence) of virulence genes, and the presence (or absence) of antibiotic resistance genes [37,38,39]. Not all enterococcal strains are the same when it comes to considering their potential pathogenicity. Modern techniques and available molecular tools make the selection of a strain without any putative virulent factors quite simple.

The aim of this study is to assess the effectiveness and safety of autoprobiotics as PFFP based on indigenous genetically tested enterococci in the early postoperative period in patients with CRC.

## 2. Materials and Methods

### 2.1. Patient’s Characteristics

A total of 36 patients diagnosed with CRC were enrolled in the study. The average age of the participants was 65.1 ± 8.9 years, with a male-to-female ratio of 2:1. Informed consent was obtained from all patients before undergoing any study procedures or therapies.

Patients were included based on main inclusion criteria: localization of CRC: left flank, age: 50–75 years, no requirement for a course of antibiotic therapy (intravenous amoxicillin on the day of surgery was permitted for preventing infectious complications). Main exclusion criteria were localization of CRC: right flank, severe cardiopulmonary pathology, cachexia.

### 2.2. Ethical Considerations

The study received approval from the Ethics Committee of the North-Western District Scientific and Clinical Center, named after L.G. Sokolov, Federal Medical and Biological Agency, Saint-Petersburg, Russian Federation. Written informed consent, following the principles outlined in the Helsinki Declaration of the World Medical Association, was obtained from all subjects. The study adhered to the “Rules of Clinical Practice in the Russian Federation” approved by the Ministry of Health of the Russian Federation (Order No. 266, 19 June 2003) and was authorized under Federal Law N323-FL dated 21 November 2011, “On Protection of Health of Citizens in the Russian Federation”. The study maintained patient confidentiality.

### 2.3. Study Design

The prospective study was performed. The first fecal sample was collected 10–14 days before surgery. Patient allocation into groups A and C, along with randomization in a 2:1 ratio (receiving autoprobiotic: without autoprobiotic, respectively), occurred before surgery. Autoprobiotic strains of *Enterococcus faecium* or *Enterococcus hirae* were isolated from fecal samples of group A patients to obtain the autoprobiotic. The study group (Group A) comprised 24 CRC patients who received autoprobiotic therapy in the early postoperative period, while the control group (Group C) included 12 CRC patients without autoprobiotic therapy. In both groups, age and the proportion of male/female subjects was the similar.

The experimental group (Group A) received autoprobiotic personalized therapy starting from the third day after surgery. Patients consumed a soy-based autoprobiotic (SUPRO PLUS 2640 DS, “Solae”, Zwaanhofweg, Belgium) containing 5 × 10^8^ CFU/mL autoprobiotic enterococci at a dose of 100 mL per day (50 mL twice a day) for 10 days. SUPRO PLUS 2640 DS, a protein–vitamin–mineral complex, is a high-quality instant dry protein product enriched with vitamin A, iron, iodine, and zinc. It has increased biological value due to the presence of soy protein and a low content of saturated fat and cholesterol [40].

The second fecal sample was collected upon completion of the autoprobiotic course.

In Group C, all examinations and analyses were performed at the same time intervals.

Prior to surgery and between days 14 and 16 post-surgery, comprehensive evaluations were conducted on all patients, encompassing the following:
Stool Analysis: Assessment of stool frequency and consistencyDigestive System and Psychoemotional Status Assessment with specialized questionnaires:a.Gastrointestinal Symptom Rating Scale (GSRS)b.Gastrointestinal Symptom Score (GIS)c.Hospital Anxiety and Depression Scale (HADS)
Examination of the gut microbiotaAnalysis of interleukins parameters

The schematic representation of the study design is depicted in Figure 1.

The study group before the surgery was designated as AV1, and after receiving the autoprobiotic treatment it was designated as AV2; the control group before the surgery was designated as CV1 and 14–16 days after surgery, as CV2.

### 2.4. Autoprobiotic Strains

The methodology for producing autoprobiotics based on non-pathogenic *E. faecium*, previously developed by our team [41], was employed. To isolate autoprobiotic strains of enterococci, we used fecal samples delivered to the laboratory no later than 2 h after defecation. Bacteria were grown on selective m Enterococcus Agar (Pronadisa, Madrid, Spain) and individual colonies were selected. All isolated cultures of autoprobiotic enterococci underwent species identification and were scrutinized for the presence of pathogenicity genes using a pre-established algorithm [42]. The identification of enterococci was conducted through polymerase chain reaction (PCR) and matrix-assisted laser desorption/ionization time-of-flight mass spectrometry (MALDI-TOF).

Enterococcal colonies were also subcultured on Colombian agar with sheep’s blood (Thermo Scientific™, Karlsruhe, Germany). The species identity of the isolates showing no hemolysis was established, and studies were conducted to identify pathogenicity genes, including surface proteins (adhesins) involved in the process of adhesion and subsequent invasion (esp, asa1, efa), as well as genes encoding gelatinase synthesis (gelE), enterococcal cytolysins (cylM, cylA), serine proteinase (sprE), and pheromone (fsrB). All *E. faecium* and *E. hirae* strains used for the autoprobiotic development were checked for the absence of the vancomycin resistance genes vanA and vanB using polymerase chain reaction (PCR). We checked that *Enterococcus* spp. strains which were chosen for making autoprobiotics had no esp, asa1, efa, gelE, cylA, cylM, sprE, fsrB in the genome.

Non-pathogenic strains of *E. faecium* and *E. hirae*, obtained from patients, were cultured in SuproPlus 2640 medium (Monsanto company, Creve Coeur, MO, USA, concentration 40 g/L), and subsequently, PFFP was prepared from these cultures.

#### Making of Autoprobiotic Enterococci

A suspension of feces in PBS at a dilution of 1:10 was plated in an amount of 100 μL on azide agar (NICF, St-Petersburg, Russia) and cultured for 48 h at a temperature of 37 °C. Mainly enterococci grew on this selective medium. Colonies typical for *E. faecium* or *E. hirae*, with a pink rim and a burgundy center, were selected. Genetic characterization of the enterococci [43,44,45,46,47] was performed employing PCR with species-specific primers and primers for identification of the virulence-related genes [48]. A single colony of a selected, identified, pure culture of *Enterococcus* spp. was added to 10 mL of the culture medium SUPRO^®^ 2649 and incubated for 48 h at 37 °C until fermentation. Then, the obtained 10 mL of starter culture was used as an inoculum to obtain 1 L of PFFP.

### 2.5. Questionnaires

The Hospital Anxiety and Depression Scale (HADS), developed in 1983 by Zigmond A.S. and Snaith R.P. [49], was employed as a straightforward and widely used method for evaluating anxiety and depression levels. Comprising 14 questions (7 each for anxiety and depression assessment), the scale requires 2–5 min for patient completion. Anxiety and depression levels are independently assessed using two subscales, where 0–7 points indicate normalcy, 8–10 points indicate subclinical expression, and 11 or more points indicate clinical expression. The maximum score for each subscale is 21 points.

The Gastrointestinal Symptom Rating Scale (GSRS), developed by the Quality of Life Department at ASTRA Hassle (Author: I. Wiklund, 1998) [50], comprises 15 questions converted into 5 scales: abdominal pain (questions 1, 4), reflux syndrome (questions 2, 3, 5), diarrhea syndrome (questions 11, 12, 14), dyspeptic syndrome (questions 6, 7, 8, 9), constipation syndrome (questions 10, 13, 15). Additionally, points are calculated on a total measurement scale (1–15 questions). The questionnaire captures complaints that have troubled the patient in the week preceding its completion. Scores for each question range from 1 to 7, with higher scores indicating more severe symptoms of gastroenterological pathology and a lower quality of life.

The Gastrointestinal Symptoms Score (GIS) comprises 10 items assessing the degree of manifestation of a broad spectrum of gastroenterological symptoms. The severity of clinical symptoms is evaluated on a five-point Likert scale from 0 to 4, where 0 represents no symptom, 1 is mild, 2 is moderate, 3 is severe, and 4 is very severe [51]. The maximum score on this scale is 40 points, with higher scores indicating more severe dyspepsia.

### 2.6. Study of the Gut Microbiota

#### 2.6.1. Bacteriological Study

Changes in the gut microbiota before and after therapy were tested by bacteriological analysis of the fecal samples using a previously described method [52]. The time intervals between collection of samples and laboratory handling did not exceed 1 h. The probes (1 g) were homogenized in 1 mL of phosphate buffered saline, PBS (8.00 g/L NaCl, 0.20 g/L KCl, 1.44 g/L Na_2_HPO_4_, 0.24 g/L KH_2_PO_4_, pH 7.4). Then the samples were diluted in 10–106 times employing method of serial dilutions. This made it possible to identify the following bacteria belonging to the genera: *Lactobacillus*, *Enterococcus*, *Bifidobacterium*, *Staphylococcus aureus*, *Escherichia*, *Proteus*, and *Klebsiella*. The following selective and differential diagnostic culture media were used for the bacteriological studies: blood agar, mannitol salt agar, MacConkey’s agar, azide agar (NICF, St-Petersburg, Russia), MRS agar (Difco, Davenport, IA, USA), and Blaurock medium (Nutrient medium, St-Petersburg, Russia). These different morphotypes were isolated and submitted to microscopic examination. Microscopic examination was conducted by way of the Gram stain procedure of pure cultures of bacteria. After enumeration of the colonies on the agar plates, three to four colonies presenting different microscopic appearances were analyzed by MALDI-TOF mass spectrometry.

#### 2.6.2. Quantitative Polymerase Chain Reaction (qPCR)

A quantitative polymerase chain reaction (qPCR) was performed using the kit “Colonoflor” (AlphaLab, Moscow, Russia), corresponding to the set of marker colonic bacteria on the qPCR, using Mini-Opticon, BioRad (Hercules, CA, USA). This methodology enables the estimation of the total bacterial count, along with the quantification of obligate and conditionally pathogenic members of the microbiota, including but not limited to *Lactobacillus* spp., *Bifidobacterium* spp., *Enterococcus* spp., *Escherichia coli*, *Escherichia coli enteropathogenic*, *Bacteroides fragilis*, *Bacteroides thetaiotaomicron*, *Faecalibacterium prausnitzii*, *Proteus mirabilis/vulgaris*, *Staphylococcus aureus*, *Klebsiella pneumoniae*, *Klebsiella oxytoca*, *Candida* spp., *Clostridioides difficile*, *Clostridium perfringens*, *Proteus* spp., *Enterobacter* spp., *Methanobrevibacter* spp., *Fusobacteria* spp., *Akkermansia* spp., *Acinetobacter* spp., *Prevotella* spp.

#### 2.6.3. Metagenome Analysis (16 S rRNA)

The metagenome analysis (16S rRNA) was performed as it was early described [48]. Fecal samples frozen on the day of the material’s collection were used for the metagenome analysis. Libraries of hypervariable regions, V3 and V4, of the 16S rRNA gene were analyzed on MiSeq (Illumina, San Diego, CA, USA). DNA were isolated from feces using the kit Express-DNA-Bio (Alkor bio, St-Petersburg, Russia). The standard method recommended by Illumina based on employing two rounds of PCR was used to prepare the libraries.

### 2.7. Serum Analysis (Cytokine Status)

The concentrations of cytokines TNF-α, IL-8, IL-10, IL-1β, IL-6, IL-18, MCP-1, and IFN-γ in the blood serum were determined using an enzyme-linked immunosorbent assay with the “Vector-Best” test system (JSC, Novosibirsk, Russia) and an ELISA analytical equipment complex (Bio-Rad, Hercules, CA, USA) according to the manufacturers’ instructions. The samples with readings below the detection limit (a concentration of 0.5 times the minimum detection value) were categorized undetectable.

### 2.8. Statistical Analysis

The Kolmogorov–Smirnov criterion was employed to assess the normality of data distribution, leading to the utilization of nonparametric criteria.

Statistically significant differences among groups were ascertained using the Mann–Whitney U-criterion, adjusted for multiple comparisons through the Benjamini–Hochberg method. Additionally, Wilcoxon’s T-test was applied for paired samples.

For comparative analysis, the post hoc test of honestly significant difference (HSD) for unequal N was utilized in the Statistica-8 software. Differences with *p* < 0.05 were deemed significant. Exploring correlations between the studied parameters involved Spearman’s test via the software package Statistica 8.0 (StatSoft, Tulsa, OK, USA).

Vectors for principal component analysis (PCA) using PERMANOVA were based on OTU abundances, filtered for noise, and normalized for total OTU counts in each sample.

## 3. Results

### 3.1. Clinical Data

Analysis of the frequency and consistency of stool before treatment showed that in patients with CRC with tumor localization in the left flank, there are stool disorders of both diarrhea (25% of patients) and constipation (25% of patients); in 50% of patients, there were no stool disorders. Analysis of the appearance of the first independent stool after surgery showed that individuals receiving an autoprobiotic had an independent stool of normal consistency (3–4 types on the Bristol scale) earlier than those who did not receive it (the stool was either 1–2 types, or 5–6 types on the Bristol scale [53,54]), and also against the background of rapid normalization of stool frequency, there was a low severity of flatulence in group A. Patients who took the treatment with autoprobiotics had a hospital stay of 5–7 days post-surgery, signifying the absence of postoperative complications.

#### 3.1.1. Analysis of the Results of the Questionnaires

Evaluation of the Hospital Anxiety and Depression Scale (HADS) scores prior to treatment indicated that, in the majority of patients, anxiety and depression levels remained below acceptable values (below 7 points), despite the established CRC diagnosis. The average anxiety level was 3 points, and the average depression level was 5.4 points. An elevated level of anxiety was identified in one patient (9 points), while depression was observed in three patients (8, 8, 10 points). After autoprobiotic intake, there was a clear tendency to decrease the level of anxiety. This may be explained by the fact that indigenous enterococci are capable of synthesizing hormones and neurotransmitters (serotonin, gamma-aminobutyric acid, etc.) [55,56].

Based on GIS questionnaire results, a reduction in dyspeptic symptoms was observed with autoprobiotic use post-surgery (Figure 2).

Furthermore, GSRS questionnaire results indicated that autoprobiotic use did not exacerbate the condition or lead to increased dyspeptic complaints after surgery (Table 1).

In the control group, any changes in GSRS dated before and after surgery were not revealed.

#### 3.1.2. Assessment of Adverse Events

During the administration of the autoprobiotic, only two episodes of adverse events were documented. One patient experienced transient nausea on the third day of use, resolving spontaneously within 2 days. In another patient, stool consistency shifted from type 4 on the Bristol scale to type 1–2, which normalized following dietary adjustments. No serious adverse events were recorded.

### 3.2. Serum Analysis (Cytokine Status)

The introduction of autoprobiotics resulted in a reduction in some pro-inflammatory cytokine levels in the blood serum (Figure 3a,b).

The cytokine levels in most patients were below normal and only exceeded it in the case of IL-18, regardless of the observation period. Perhaps this was due to the presence of immunological refractoriness after the inflammatory reaction or immuno-pathological processes. Statistically significant changes in all cytokine concentrations in the blood serum before and after treatment in group C were not revealed. The accent was placed on the analysis of the content of pro-inflammatory and anti-inflammatory cytokines in the blood serum in order to assess how the intake of autoprobiotics influences the cytokine balance. As a result, we did not find a significant increase for most cytokines, which may be explained by the changes in immunity of the patients with CRC after PFFP usage. Statistically significant changes were demonstrated only for IL-6 and IL-18. A decrease in IL-18 and IL-6 concentrations in the blood serum after autoprobiotic consumption may also be associated with a decrease in the inflammatory response by suppressing the reproduction of pathogenic microorganisms and microbiota restoration.

### 3.3. Cut Microbiota Study

#### 3.3.1. Bacteriological Study

The bacteriological study was mainly aimed to isolate non-pathogenic *Enterococcus* spp. for autoprobiotic production. Genetic analysis of DNA from enterococci grown on a selective medium (azide agar) facilitated species identification and detection of pathogenicity and vancomycin resistance genes in the genome. Patients (24 individuals) from whom non-pathogenic *E. faecium* and/or *E. hirae* were isolated from feces were assigned to group A. *E. faecalis* were also isolated from fecal samples of patients with CRC.

When assessing the content of Enterococcus spp. in the stool after taking autoprobiotics, we were able to monitor a decrease in the abundance of *E. faecalis* and an increase in *E. faecium* or *E. hirae*. Alterations in the abundance of various enterococci species are depicted in Figure 4: a statistically significant decrease in *E. faecalis* (can induce excessive collagen degradation [57]) (Figure 4A) and an increase in non-pathogenic *E. faecium* or *E. hirae* was found (Figure 4B).

Furthermore, in the bacteriological analysis following the administration of autoprobiotics, a shift in escherichia populations was observed, transitioning from atypical (lactase-negative, hemolytic, sucrose-positive, and displaying low enzymatic activity) to typical forms (Figure 5).

Additionally, there was a tendency toward an increase in the abundance of bifidobacteria, which are obligate beneficial members of the intestinal microbiota (Figure 6).

#### 3.3.2. qPCR Study Results

When analyzing the composition of the gut microbiota before and after autoprobiotic usage in the early post-surgery period, there were no negative changes in people receiving an autoprobiotic, despite the appearance in the lives of patients of several factors contributing to the deterioration of the gut condition (surgery). After autoprobiotic intake, we saw an increase in the *Lactobacillus* spp. and *E. faecium* quantity, a decrease in the *Akkermantia muciniphyla* and *Clostridium perfringents* populations, disappearance of the microbial cancer marker *Parvimonas micra*, and the tendency for a decrease in the content of *Enterobacter* spp. (Figure 7).

An important positive change after autoprobiotic therapy is the disappearance of *P. micra* and *Fusobacterium nucleatum*, microorganisms that may be associated with the progression of CRC [58,59,60,61,62,63,64,65]. A decrease in the quantitative content of akkermansia contributes not only to the stabilization of the gut mucosa, but also to a decrease in the level of some interleukin, which we revealed during therapy [66,67]. An increase in *C. perfringens* content is a characteristic feature of the microbiota in CRC, and a decrease in their content after therapy with autoprobiotics indicates its success.

#### 3.3.3. Metagenome Analysis Study

Using metagenome analysis, we found statistically significant differences before and after therapy in only the alpha biodiversity in the class level (*p* = 0.023) and a tendency in other levels (Figure 8 and Appendix A). It can be considered as a sign of microbiota recovery.

## 4. Discussion

This paper presents a prospective study of the effect of autoprobiotics on the human body and the microbiota in the postoperative period when patients may have a large number of complications of surgical intervention. First of all, these complications manifest in dyspepsia, the occurrence of inflammatory reactions, and even greater disorders of the gut microbiota [68]. In previous experiments, other authors have proven the possibility of using probiotics in the complex therapy of CRC [69]. For the first time, we used autoprobiotics for CRC therapy. They are characterized by greater safety (autoprobiotics can be stored in the patient’s body for a longer time without entering into conflict with the microbiota, they are protected by the immune system with immunological tolerance, and they adapt to the conditions of coexistence as part of a consortium of microorganisms and habitual living conditions) [30]. Previously, we conducted a study on the effect of autoprobiotic strains of *E. faecium*, bifidobacteria, lactobacilli, and their mixtures on the model of experimental antibiotic-associated intestinal dysbiosis in Wistar rats. We found a positive effect, primarily of enterococci and bifidobacteria, in correcting dysbiosis manifested by the excessive growth of opportunistic representatives of the *Enterobacteriaceae* family. The correction of dysbiosis was accompanied by a stimulation of anti-inflammatory cytokines IL-10, TGF-β, and a reduction in proinflammatory cytokines MCP1 and IL-8 [35]. The same model shows a beneficial effect of autoprobiotics on intestinal motility [70]. Also, we demonstrated that autoprobiotic intake contributes to a decrease in dyspeptic symptoms and is associated with positive changes in the gut microbiota content [71,72], as well as the vaginal microbiota content [73].

Major mechanisms contributing to neoplastic transformation involve disruptions in gut microbiota, synthesis of “pathological” metabolites, and the production of genotoxins by gastrointestinal bacteria. These factors lead to immune dysregulation and inflammation [74,75]. Pro-inflammatory cytokines and hormones play a significant role in CRC pathogenesis. Excessive human chorionic gonadotropin, a negative prognostic factor for CRC, stimulates TNFα and IL-6 hypersecretion, as well as the synthesis of IL-8 and matrix metalloproteinases (MMPs), facilitating the invasion of microorganisms beyond the gut mucosa [76,77,78]. Studies have reported negative prognoses for CRC with elevated IL-6 and TNF-α levels in peritoneal fluid during the postoperative period [79]. Other authors also observed an elevation in IL-6 and TNF-α levels in the blood serum of CRC patients, with a subsequent decrease post-surgery by 40–60% [76]. In this respect, the decrease in pro-inflammatory signaling after the consumption of autoprobiotics can be considered positive. Indeed, the levels of two important proinflammatory cytokines (IL-6 and IL-18) decreased, while all the rest of the cytokines under study remained the same.

A special feature of this study was the additional use of *E. hirae*, which we first discovered as candidates for autoprobiotic agents in Vietnam [80,81], and after that, in Russia for patients with CRC.

The choice of the genus enterococci was very successful, as they displaced pathogenic enterococci more often. Antagonism between closely related taxa usually manifests itself to a greater extent; in this study, there is a potential antagonism between *E. faecium*/*E. hirae* and *E. faecalis* that cannot be excluded. The effect of autoprobiotic enterococci on humans has several points of action: directly by its metabolites and indirectly through the modulation of the gut microbiota (pathogenic and oncogenic bacteria are leaving; useful ones may increase). The role of bacterial taxa is also not completely clear, as evidenced by the huge number of reports on the duplicitous role of *Akkermansia muciniphila*, *Bifidobacterium* spp., *Bacteroides* spp., etc. [6,7].

As a result of this study, we have identified a positive effect of autoprobiotics in the early postoperative period of CRC. During and after intake of non-pathogenic indigenous enterococci, we did not see any serious adverse events, which confirms their safety. It is important that the addition of autoprobiotics contributed to a significant decrease in dyspepsia and a decrease in the number of postoperative complications, which improves the quality of life of patients. Patients receiving autoprobiotics, compared to the control group, reported fewer postoperative complaints, as validated by the GIS for dyspepsia assessment and the GSRS for gastroenterological complaints. Improving the clinical data and stool frequency and form correspond to the results described when using probiotics by other authors in the treatment of oncological diseases [53,54]. Positive changes by the GSRS questionnaire in group A patients underscore the high efficacy of autoprobiotics in preventing postoperative stool disorders and dyspepsia in CRC patients. Absence of any changes by the HADS suggests that the psycho-emotional factor may not be a primary contributor to CRC development, and the short period post-diagnosis may not significantly influence the patient’s psychological well-being. Autoprobiotic usage in the therapy of CRC led to an improvement in the quality of life, which is extremely important for this category of patients. In particular, a clearly identified tendency to decrease anxiety may be explained by the fact that indigenous enterococci are capable of synthesizing hormones and neurotransmitters (serotonin, gamma-aminobutyric acid, etc.), which positively affect the central nervous system, reduce the severity of anxiety and depressive disorders, and improve the quality of life [55,56].

The use of autoprobiotics led to positive changes in the gut microbiota content, including a decrease in the number of atypical *Escherichia coli*, in the quantitative content of *C. perfringens*, an increase in *E. faecium*/*E. hirae*, a decrease in more pathogenic *E. faecalis*, and an increase in alpha diversity.

A decrease in *E. faecalis* levels and an elevation in the prevalence of generally non-pathogenic enterococci, specifically *E. faecium* and *E. hirae*, can be regarded as favorable changes. Primarily, this is attributed to the fact that *E. faecalis* has the capability to bind and locally activate human fibrinolytic protease plasminogen (PLG). The activation of PLG by *E. faecalis* induces excessive collagen degradation [57]. Notably, *E. faecalis* is the pathogen most frequently identified in postsurgical colonic perforations in humans.

The disappearance of *P. micra* and *F. nucleatum* after autoprobiotic intake is an important positive change for patients with CRC.

It is well known that *F. nucleatum* is part of the commensal flora of the intestine and oral cavity, but its presence has been associated with pathological conditions including appendicitis, inflammatory bowel disease (IBD), periodontitis, and CRC. Experimental biological models have demonstrated many mechanisms by which *F. nucleatum* can contribute to the progression of CRC, including E-cadherin-mediated activation of Wnt/β-catenin signaling.15 [58]. It has also been suggested that *F. nucleatum* inhibits the antitumor immune response [59] and decreases the effect of chemotherapy [60].

*P. micra*, like *F. nucleatum*, is a commensal of the oral cavity and participates in the pathogenesis of intracranial abscess, pericarditis, and necrotizing fasciitis, as well as CRC [61,62,63,64]. However, the role of *P. micra* in the progression of CRC is still largely unknown, and the potential of these bacteria as a fecal marker for detecting CRC has not been fully elucidated [65]. In this study, *P. micra* was completely eliminated after exposure to an autoprobiotic, which can be considered as one of the positive effects of complex therapy.

A decrease in the quantitative content of akkermansia contributes not only to the stabilization of the gut mucosa, but also to a decrease in the level of some interleukin, which we identified during therapy. According to a meta-analysis, *Akkermansia* spp. populations tend to increase in CRC; however, there are publications indicating the opposite [6]. In addition, it should be taken into account that mucin degradation caused by *A. muciniphila* may reduce the thickness of the mucin layer and increase the risk of infectious complications in the digestive tract [66]. Moreover, *A. muciniphila* protein (named Amuc_1100 protein) interacting with Toll-like receptor 2 (TLR2) could induce a wide range of immunomodulatory responses, including the production of cytokines IL-6, IL-8, and IL-10 [67].

A decrease in the concentration of two pro-inflammatory cytokines can be associated with positive changes of the gut microbiota composition. Interpretation of these findings is complex, as there is limited prior observation of changes in cytokine status in CRC patients.

Future studies will allow the discovery of additional fine mechanisms of autoprobiotic therapy and its impact on the digestive, immune, endocrine, and neural systems.

## 5. Conclusions

Autoprobiotics present a promising avenue for complementary therapy of CRC. The administration of autoprobiotics in the postoperative period is highly effective and safe in the comprehensive treatment of CRC patients, offering:A personalized approach to patient care.Expedited restoration of stool of normal consistency after surgery (type 3–4 on the Bristol scale).Reduction in abdominal pain, dyspeptic symptoms, and inflammation post-surgery.Lowering the risk of CRC relapse by diminishing pro-carcinogenic inflammation in the colon associated with gut dysbiosis.A complex approach in the restoring of composition of the gut microbiota in complex therapy of CRC after surgical intervention with autoprobiotic enterococci.

Postoperative use of autoprobiotics facilitates the postoperative course, enhances the clinical outlook, mitigates inflammation, and supports the restoration of gut microbiocenosis and its functions.

## Figures and Tables

**Figure 1 microorganisms-12-00980-f001:**
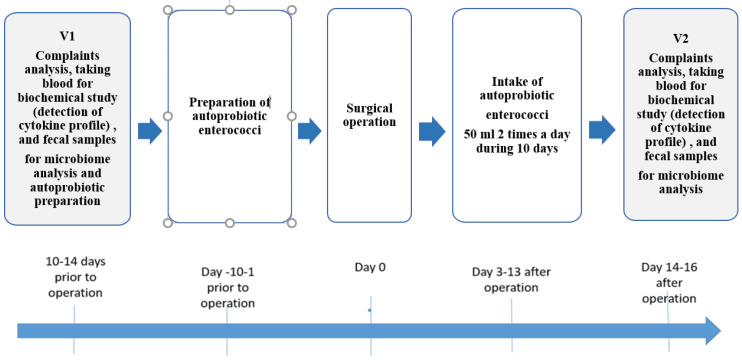
Study design.

**Figure 2 microorganisms-12-00980-f002:**
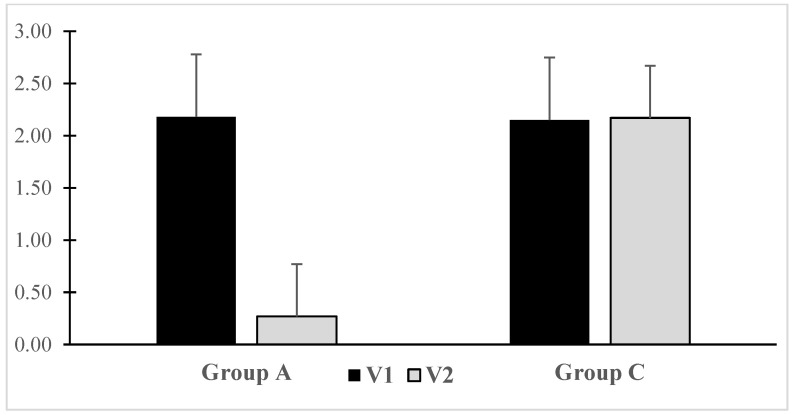
Dynamics of dyspepsia severity according to the GIS scale in patients with CRC after intake of indigenous enterococci. Notes: V1—before autoprobiotic consumption, V2—after autoprobiotic consumption.

**Figure 3 microorganisms-12-00980-f003:**
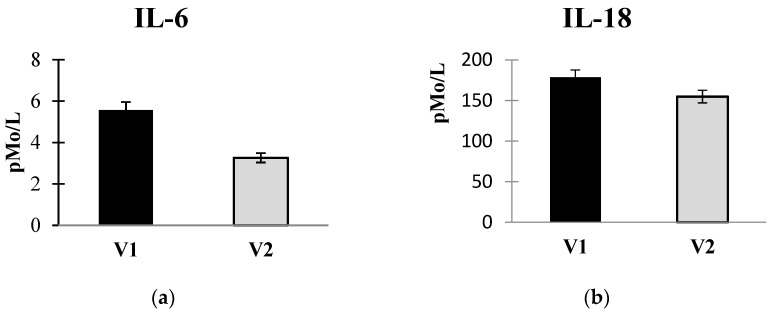
Changes in serum levels of IL-6 (**a**) and IL-18 (**b**) before and after autoprobiotic administration. Note: Data are presented in pmol per 1 L. V1—before autoprobiotic consumption, V2—after autoprobiotic consumption. The upper norm limit for IL-6 is 7 pmol per 1 L, and for IL-18 it is 100 pmol per 1 L.

**Figure 4 microorganisms-12-00980-f004:**
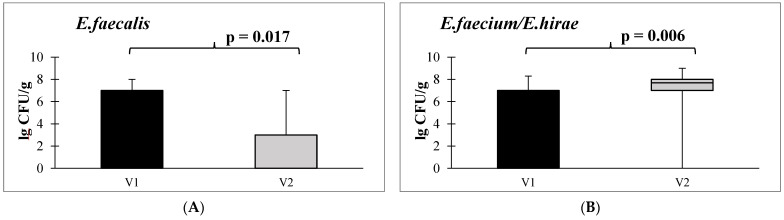
Quantitative content of *E. faecalis* (**A**) and *E. faecium* or *E. hirae* (**B**) before and after administration of autoprobiotics in the early postoperative period. Note: V1—before autoprobiotic consumption, V2—after autoprobiotic consumption.

**Figure 5 microorganisms-12-00980-f005:**
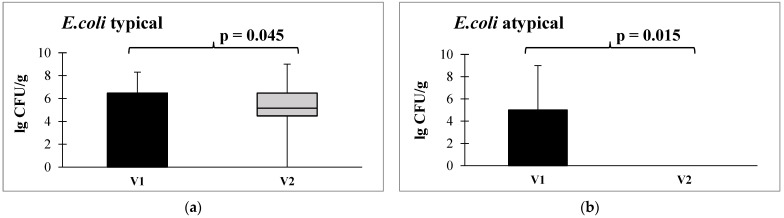
Quantitative content of atypical *E. coli* (**a**) and typical *E. coli* (**b**) before and after administration of autoprobiotics in the postoperative period. Note: V1—before autoprobiotic consumption, V2—after autoprobiotic consumption.

**Figure 6 microorganisms-12-00980-f006:**
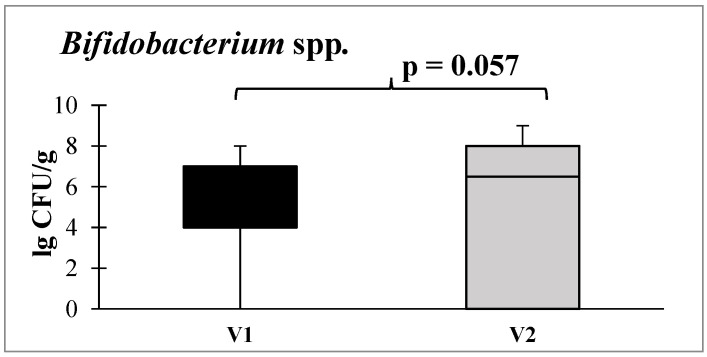
Quantitative content of *Bifidobacterium* spp. before and after administration of autoprobiotics in the postoperative period. Note: V1—before autoprobiotic consumption, V2—after autoprobiotic consumption.

**Figure 7 microorganisms-12-00980-f007:**
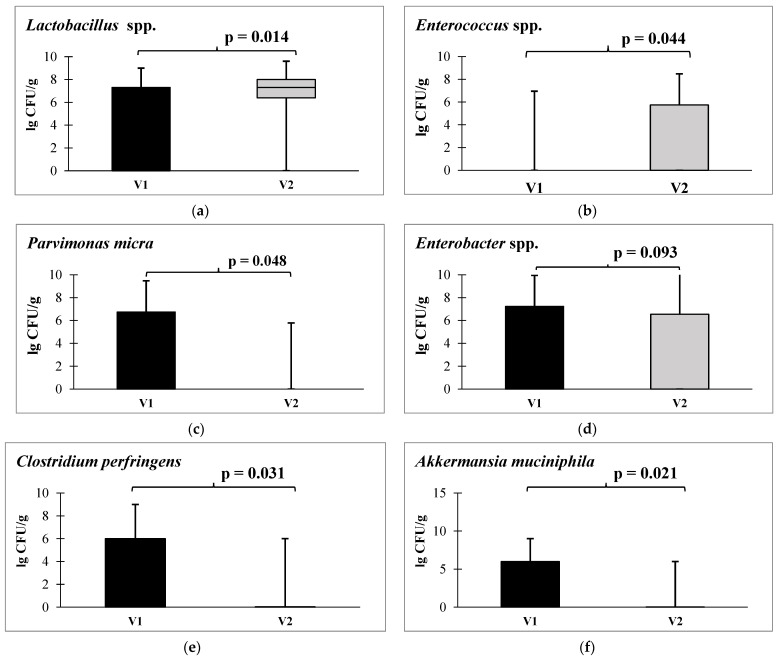
*Lactobacillus* spp. (**a**), *Enterococcus* spp. (**b**), *Parvimonas micra* (**c**), *Enterobacter* spp. (**d**), *Clostridium perfringens* (**e**), and *Akkermansia muciniphila* (**f**) quantitative content in the fecal samples of patients with CRC before and after therapy. Note: V1—before autoprobiotic consumption, V2—after autoprobiotic consumption.

**Figure 8 microorganisms-12-00980-f008:**
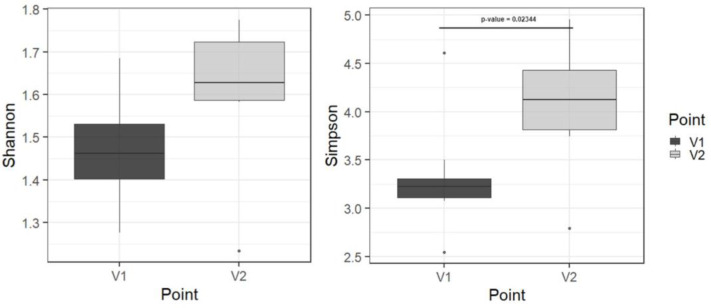
Alpha-biodiversity before and after therapy in the level of class. Note: V1—before autoprobiotic consumption, V2—after autoprobiotic consumption.

**Table 1 microorganisms-12-00980-t001:** The average results of the GSRS questionnaire in dynamics before and after autoprobiotic intake.

Parameters, Scores/Data Evaluation Periods	Abdominal Pain	Reflux	Diarrhea	Dyspepsia	Constipation
V1	1	1	3.5	2.4	1.1
V2	0	0	1.1 *	0.75 *	0

Notes: * *p* < 0.05. V1—before autoprobiotic consumption, V2—after autoprobiotic consumption.

## Data Availability

Data are contained within the article and Appendix A.

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
