# Peer review of "Autoprobiotics in the Treatment of Patients with Colorectal Cancer in the Early Postoperative Period"

_microorganisms, 2024, doi:10.3390/microorganisms12050980_

Round 1
Reviewer 1 Report
Comments and Suggestions for Authors
The authors of this study evaluated the effectiveness of the use of autoprobiotics based on indigenous non-pathogenic strains of Enterococcus faecium and Enterococcus hirae in the complex therapy of colorectal cancer (CRC) in the early postoperative period in 36 patients diagnosed with CRC. The study group (Group A) comprised 24 CRC patients who received autoprobiotic therapy in the early postoperative period, while the control group (Group C) included 12 CRC patients without autoprobiotic therapy. The authors evaluated the included patients before surgery and up to day 14-16 post-surgery through stool and gastroenterological complains analysis, examination of the gut microbiota (bacteriological study, quantitative polymerase chain reaction, metagenome analysis), and analysis of interleukins in serum. The authors found that autoprobiotics decreased dyspeptic complaints after surgery with no postoperative complications or any side effects, and decreased decreased pro-inflammatory cytokines (IL-6 and IL-18). Moreover, the use of autoprobiotics led to positive changes in the structure of escherichia and enterococci populations, the elimination of Parvomonas micra, Fusobacterium nucleatum, decrease in the quantitative content of Clostridium perfringens, Akkermansia muciniphila. Metagenomic analysis (16S rRNA) revealed an increase in alpha diversity. The authors concluded that administration of autoprobiotics in the postoperative period is highly effective and safe in the comprehensive treatment of colorectal cancer patients.
This is a nice study with an interesting topic.
The authors should state the nature of the study i.e., retrospective study or prospective study.
“Main un-inclusion criteria”. Please rephrase as “exclusion criteria”
Author Response
Dear reviewer. Thank you for your revision. It is very important and help us to improve the quality of our work. Please find the answers on your questions:
- The authors should indicate the nature of the study, i.e. a retrospective study or a prospective analysis. Answer: We made the corrections
- “The main criteria for non-inclusion". Please rephrase as “exclusion criteria". Answer: We made the corrections
Reviewer 2 Report
Comments and Suggestions for Authors
Author Response
Dear reviewer
Thank you for your revision. It is very important and help us to improve the quality of our work
Please find the answers on your questions:
- While I appreciate the effort of the work presented and the significance of comparing Enterococcus strains usage results with intestinal microflora analysis, I think the authors needs to improve the focus of the paper and provide more information on the outcome of the research.
Answer:
We add more information about study results.
- Please correct the text of theses. Latin names are written in Italic font. Please improve throughout your work
Answer:
We made the corrections
- The manuscript appears to be reporting some significant measurements made on the intestinal microflora analysis, however, the impact is lost by a seemingly short discussion of the findings Answer:
We have increased the discussion. The discussion in this article is partially included in the research results section. In addition, fragments of the discussion were added to the discussion section summarizing the results.
- I have serious concerns over the use of Enterococcus strains isolated from human feces. I believe that Enterococcus strains usually have transposons holding antibiotic resistant genes. Have you ever confirmed the existence of the antibiotic-resistant genes in these strains?
Answer:
We add information that confirm the existence of the antibiotic-resistance genes in these strains in our study. We have added additional comments and judgments confirming the importance of choosing enterococcal strains for the autoprobiotic making for the usage as personalized functional food product (PFFP) in introduction part.
- You need to justify why Enterococcus was chosen and not another GRAS bacteria. Confirm that Enterococcal strains had not had virulence genes.
Answer:
There is a lot of information about pathogenic enterococci and the successful use of probiotic products and -medicines based on non parthogenic enterococcal strains, isolated from natural functional food products or human microbiota in medicine. Previously, this information were summarized in our co-author Suvorov A. rereview (doi: 10.3920/BM2017.0148. Epub 2018 Apr 10. PMID: 29633645. doi: 10.1007/s12602-020-09633-y. PMID: 31955388).
We add information why we choose Enterococcus spp. and not another GRAS bacteria. We chose Enterococcus hirae and E. faecium not another generally recognized as safe (GRAS) bacteria because they were more effective in our previous experiments, in particular when comparing the effects of endogenous lactobacilli, bifidobacteria and their mixtures on models of experimental dysbiosis and on cell culture in assessing the effect on immunity. In addition, these bacteria are biotechnological and are easily cultivated and do not die as quickly as bifidobacteria and lactobacilli.
Genetic studies of enterococci revealed that probiotic strains selected for human consumption are quite different from clinical isolates by the organization of their genomes, the presence (or absence) of virulence genes, and the presence (or absence) of antibiotic resistance genes. Conjugative gene transfer in enterococci requires special conditions, certain recipient strains and donors with conjugative plasmids, quorum sensing signaling, and a certain ratio between donors and recipients. Besides, several GRAS Lactobacillus probiotics are also vancomycin-resistant and carry a vancomycin resistance regulon in their genome, which is euphemistically considered to be an intrinsic resistance factor. Not all enterococcal strains are the same when it comes to considering their potential pathogenicity. Modern techniques and available molecular tools make the selection of a strain without any putative virulent factors quite simple. In our study all E. faecium and E. hirae strains used for the autoprobiotic making were checked for the absence of the vancomycin resistance genes vanA and vanB using polymerase chainreaction (PCR). We checked that Enterococcus spp. strains which were chosen for making autoprobiotics have not esp, asa1, efa, gelE, cylA, cylM, sprE, fsrB in the genome.
- About proinflammatory cytokines, the authors write in very general terms. Please provide details of the serum analysis
Answer:
We included more details of the serum analysis method, result of immunological study and their discussion in the manuscript.
- Prepare graphs properly. Axis titles are overlapped in some places, and not displayed fully at several graphs. The size of all graphs should be adjusted
Answer:
All graphs were corrected.
- Adjust the size of 2 graphs in Figure 8.
Answer:
All the figures including figure 8 have been redone. The size of all graphs was adjusted.

Round 2
Reviewer 2 Report
Comments and Suggestions for Authors
The revised manuscript has improved clearly and, the authors have addressed carefully most comments I raised previously. I have some additional comments.
1. Please, check carefully the formats of the scientific names. Still, there are several places with non-Italic formats.
2. Unfortunately, the writing of the Results and Discussion sections is imprecise. These sections usually highlight your findings and the most important parts of the manuscript. However, in the Results section, you are still discussing the findings!? I should recommend to re-organize these sections.
Author Response
Dear reviewer
Thank you for your revision. It is very important and help us to improve the quality of our work
Please find the answers on your questions:
- Please, check carefully the formats of the scientific names. Still, there are several places with non-Italic formats.
Answer:
We made the corrections
- Unfortunately, the writing of the Results and Discussion sections is imprecise. These sections usually highlight your findings and the most important parts of the manuscript. However, in the Results section, you are still discussing the findings!? I should recommend to reorganize these sections.
Answer:
We reorganized sections results and discussion